# Optimistic policy iteration and natural actor-critic: A unifying view and a non-optimality result

**Paul Wagner**
Department of Information and Computer Science
Aalto University
FI-00076 Aalto, Finland
`paul.wagner@aalto.fi`

## Abstract

Approximate dynamic programming approaches to the reinforcement learning problem are often categorized into greedy value function methods and value-based policy gradient methods. As our first main result, we show that an important subset of the latter methodology is, in fact, a limiting special case of a general formulation of the former methodology; optimistic policy iteration encompasses not only most of the greedy value function methods but also natural actor-critic methods, and permits one to directly interpolate between them. The resulting continuum adjusts the strength of the Markov assumption in policy improvement and, as such, can be seen as dual in spirit to the continuum in TD($\lambda$)-style algorithms in policy evaluation. As our second main result, we show for a substantial subset of soft-greedy value function approaches that, while having the potential to avoid policy oscillation and policy chattering, this subset can never converge toward an optimal policy, except in a certain pathological case. Consequently, in the context of approximations (either in state estimation or in value function representation), the majority of greedy value function methods seem to be deemed to suffer either from the risk of oscillation/chattering or from the presence of systematic sub-optimality.

## 1 Introduction

We consider the reinforcement learning problem in which one attempts to find an approximately optimal policy for controlling a stochastic nonlinear dynamical system. We focus on the setting in which the target system is actively sampled during the learning process. Here the sampling policy changes during the learning process in a manner that depends on the main policy being optimized. This learning setting is often called interactive learning [e.g., 23, §3]. Many approaches to the problem are value-based and build on the methodology of simulation-based approximate dynamic programming [23, 4, 9, 19, 8, 21]. The majority of these methods are often categorized into greedy value function methods (critic-only) and value-based policy gradient methods (actor-critic) [e.g., 23, 13].

Within this interactive setting, the policy gradient approach has better convergence guarantees, with the strongest case being for Monte Carlo evaluation with 'compatible' value function approximation. In this case, convergence with probability one (w.p.1) to a local optimum can be established for arbitrary differentiable policy classes under mild assumptions [22, 13, 19]. On the other hand, while the greedy value function approach is often considered to possess practical advantages in terms of convergence speed and representational flexibility, its behavior in the proximity of an optimum is currently not well understood. It is well known that interactively operated approximate hard-greedy

---

An extended version of this paper with full proofs and additional background material is available at `http://books.nips.cc/` and `http://users.ics.aalto.fi/pwagner/`.

value function methods can fail to converge to any single policy and instead become trapped in sustained policy oscillation or policy chattering, which is currently a poorly understood phenomenon [6, 7]. This applies to both non-optimistic and optimistic policy iteration (value iteration being a special case of the latter). In general, the best guarantees for this methodology exist in the form of sub-optimality bounds [6, 7]. The practical value of these bounds, however, is under question (e.g., [2; 7, §6.2.2]), as they can permit very bad solutions. Furthermore, it has been shown that these bounds are tight [7, §6.2.3; 12, §3.2].

A hard-greedy policy is a discontinuous function of its parameters, which has been identified as a key source of problems [18, 10, 17, 22]. In addition to the observation that the class of stochastic policies may often permit much simpler solutions [cf. 20], it is known that continuously stochastic policies can also re-gain convergence: both non-optimistic and optimistic soft-greedy approximate policy iteration using, for example, the Gibbs/Boltzmann policy class, is known to converge with enough softness, 'enough' being problem-specific. This has been shown by Perkins & Precup [18] and Melo et al. [14], respectively, although with no consideration of the quality of the obtained solutions nor with an interpretation of how 'enough' relates to the problem at hand. Unfortunately, the aforementioned sub-optimality bounds are also lost in this case (consider temperature $\tau \to \infty$); while convergence is re-gained, the properties of the obtained solutions are rather unknown.

To summarize, there are considerable shortcomings in the current understanding of the learning dynamics at the very heart of the approximate dynamic programming methodology. We share the belief of Bertsekas [5, 6], expressed in the context of the policy oscillation phenomenon, that a better understanding of these issues "has the potential to alter in fundamental ways our thinking about approximate DP."

In this paper, we provide insight into the convergence behavior and optimality of the generalized optimistic form of the greedy value function methodology by reflecting it against the policy gradient approach. While these two approaches are considered in the literature mostly separately, we are motivated by the belief that it is eventually possible to fully unify them, so as to have the benefits and insights from both in a single framework with no artificial (or historical) boundaries, and that such a unification can eventually resolve the issues outlined above. These issues revolve mainly around the greedy methodology, while at the same time, solid convergence results exist for the policy gradient methodology; connecting these methodologies more firmly might well lead to a fuller understanding of both.

After providing background in Section 2, we take the following steps in this direction. First, we show that natural actor-critic methods from the policy gradient side are, in fact, a limiting special case of optimistic policy iteration (Sec. 3). Second, we show that while having the potential to avoid policy oscillation and chattering, a substantial subset of soft-greedy value function approaches can never converge to an optimal policy, except in a certain pathological case (Sec. 4). We then conclude with a discussion in a broader context and use the results to complete a high-level convergence and optimality property map of the variants of the considered methodology (Sec. 5).

## 2  Background

A Markov decision process (MDP) is defined by a tuple $\mathcal{M} = (\mathcal{S}, \mathcal{A}, \mathcal{P}, r)$, where $\mathcal{S}$ and $\mathcal{A}$ denote the state and action spaces. $S_t \in \mathcal{S}$ and $A_t \in \mathcal{A}$ denote random variables at time $t$. $s, s' \in \mathcal{S}$ and $a, b \in \mathcal{A}$ denote state and action instances. $\mathcal{P}(s, a, s') = \mathbb{P}(S_{t+1} = s'|S_t = s, A_t = a)$ defines the transition dynamics and $r(s, a) \in \mathbb{R}$ defines the expected immediate reward function. Non-Markovian aggregate states, i.e., subsets of $\mathcal{S}$, are denoted by $y$. A policy $\pi(a|s, \theta_k) \in \Pi$ is a stochastic mapping from states to actions, parameterized by $\theta_k \in \Theta$. Improvement is performed with respect to the performance metric $J(\theta) = 1/H \sum_t^H \mathbb{E}[r(S_t, A_t)|\pi(\theta)]$. $\nabla_\theta J(\theta_k) \in \Theta$ denotes a parameter gradient at $\theta_k$. $\nabla_\pi J(\theta_k) \in \Pi$ denotes the corresponding policy gradient in the selected policy space. We define the policy distance $\|\pi_u - \pi_v\|$ as some $p$-norm of the action probability differences $(\sum_s \sum_a |\pi_u(a|s) - \pi_v(a|s)|^p)^{1/p}$. Action value functions $\bar{Q}(s, a, \hat{w}_k)$ and $Q(s, a, \hat{w}_k)$, parameterized by $\hat{w}_k$, are estimators of the $\gamma$-discounted cumulative reward $\sum_t \gamma^t \mathbb{E}[r(S_t, A_t)|S_0 = s, A_0 = a, \pi(\theta_k)]$ for some $(s, a)$ when following some policy $\pi(\theta_k)$. The state value function $V(s, \hat{w}_k)$ is an estimator of such cumulative reward that follows some $s$. We use $\epsilon$ to denote a small positive infinitesimal quantity.

We focus on the Gibbs (Boltzmann) policy class with a linear combination of basis functions $\phi$:

$$\pi(a|s, \theta_k) = \frac{e^{\theta_k^\top \phi(s,a)}}{\sum_b e^{\theta_k^\top \phi(s,b)}} \ . \tag{1}$$

We shall use the term 'semi-uniformly stochastic policy' for referring to a policy for which $\pi(a|s) = c_s \vee \pi(a|s) = 0, \ \forall s, a, \ \forall s \ \exists c_s \in [0,1]$. Note that both the uniformly stochastic policy and all deterministic policies are special cases of semi-uniformly stochastic policies.

For the value function, we focus on least-squares linear-in-parameters approximation with the same basis $\phi$ as in (1). We consider both advantage values [see 22, 19]

$$\bar{Q}_k(s, a, \hat{w}_k) = \hat{w}_k^\top \left( \phi(s,a) - \sum_b \pi(b|s, \theta_k)\phi(s,b) \right) \tag{2}$$

and absolute action values

$$Q_k(s, a, \hat{w}_k) = \hat{w}_k^\top \phi(s,a) \ . \tag{3}$$

Evaluation can be based on either Monte Carlo or temporal difference estimation. We focus on optimistic policy iteration, which contains both non-optimistic policy iteration and value iteration as special cases, and on the policy gradient counterparts of these.

In the general form of optimistic approximate policy iteration (e.g., [7, §6.4]; see also [6, §3.3]), a value function parameter vector $w$ is gradually interpolated toward the most recent evaluation $\hat{w}$:

$$w_{k+1} = w_k + \kappa_k(\hat{w}_k - w_k) \ , \quad \kappa_k \in (0,1] \ . \tag{4}$$

Non-optimistic policy iteration is obtained with $\kappa_k = 1, \ \forall k$ and 'complete' evaluations $\hat{w}_k$ (see below). The corresponding Gibbs soft-greedy policy is obtained by combining (1) and a temperature (softness) parameter $\tau$ with

$$\theta_{k+1} = w_{k+1}/\tau_k \ , \quad \tau_k \in (0, \infty) \ . \tag{5}$$

Hard-greedy iteration is obtained in the limit as $\tau \to 0$.

In optimistic policy iteration, policy improvement is based on an incomplete evaluation. We distinguish between two dimensions of completeness, which are evaluation *depth* and evaluation *accuracy*. By evaluation depth, we refer to the look-ahead depth after which truncation with the previous value function estimate occurs. For example, LSPE(0) and LSTD(0) [e.g., 15] implement shallow and deep evaluation, respectively. With shallow evaluation, the current value function parameter vector $w_k$ is required for look-ahead truncation when computing $\hat{w}_{k+1}$. Inaccurate (noisy) evaluation necessitates additional caution in the policy improvement process and is the usual motivation for using (4) with $\kappa < 1$.

It is well known that greedy policy iteration can be non-convergent under approximations [4]. The widely used projected equation approach can manifest convergence behavior that is complex and not well understood, including bounded but potentially severe sustained policy oscillations [6, 7] (see the extended version for further details). Similar consequences arise in the context of partial observability for approximate or incomplete state estimation [e.g., 20, 16]. A novel explanation to the phenomenon in the non-optimistic case was recently proposed in [24, 25], where policy oscillation was re-cast as sustained overshooting over an attractive stochastic policy. Policy convergence can be established under various restrictions (see the extended version for further details). Most importantly to this paper, convergence can be established with continuously soft-greedy action selection [18, 14], in which case, however, the quality of the obtained solutions is unknown.

In policy gradient reinforcement learning [22, 13, 19, 8], improvement is obtained via stochastic gradient ascent:

$$\theta_{k+1} = \theta_k + \alpha_k G(\theta_k)^{-1} \frac{\partial J(\theta_k)}{\partial \theta} = \theta_k + \alpha_k \eta_k \ , \tag{6}$$

where $\alpha_k \in (0, \infty)$, $G$ is a Riemannian metric tensor that ideally encodes the curvature of the policy parameterization, and $\eta_k$ is some estimate of the gradient. With value-based policy gradient methods, using (1) together with either (2) or (3) fulfills the 'compatibility condition' [22, 13]. With (2), the value function parameter vector $\hat{w}_k$ becomes the natural gradient estimate for the evaluated policy $\pi(\theta_k)$, leading to natural actor-critic algorithms [11, 19], for which

$$\eta_k = \hat{w}_k \ . \tag{7}$$

For policy gradient learning with a 'compatible' value function and Monte Carlo evaluation, convergence w.p.1 to a local optimum is established under standard assumptions [22, 13]. Temporal difference evaluation can lead to sub-optimal results with a known sub-optimality bound [13, 8].

## 3   Forgetful natural actor-critic

In this section, we show that an important subset of natural actor-critic algorithms is a limiting special case of optimistic policy iteration. A related connection was recently shown in [24, 25], where a modified form of the natural actor-critic algorithm by Peters & Schaal [19] was shown to correspond to non-optimistic policy iteration. In the following, we generalize and simplify this result: by starting from the more general setting of optimistic policy iteration, we arrive at a unifying view that both encompasses a broader range of greedy methods and permits interpolation between the approaches directly with existing (unmodified) methodology.

We consider the Gibbs policy class from (1) and the linear-in-parameters advantage function from (2), which form a 'compatible' actor-critic setup. We assume deep policy evaluation (cf. Section 2). We begin with the natural actor-critic (NAC) algorithm by Peters & Schaal [19] (cf. (6) and (7)) and generalize it by adding a forgetting term:

$$\theta_{k+1} = \theta_k + \alpha_k \eta_k - \kappa_k \theta_k \; , \tag{8}$$

where $\alpha_k \in (0, \infty)$, $\kappa_k \in (0, 1]$. We refer to this generalized algorithm as the forgetful natural actor-critic algorithm, or NAC($\kappa$). In the following, we show that this algorithm is, within the discussed context, equivalent to the general form of optimistic policy iteration in (4) and (5), with the following translation of the parameterization:

$$\tau_k = \frac{\kappa_k}{\alpha_k} \; , \quad \text{or} \quad \alpha_k = \frac{\kappa_k}{\tau_k} \; . \tag{9}$$

Taking the forgetting factor $\kappa$ in (8) toward zero leads back toward the original natural actor-critic algorithm, with the implication that the original algorithm is a limiting special case of optimistic policy iteration.

**Theorem 1.** *For the case of deep policy evaluation (Section 2), the natural actor-critic algorithm for the Gibbs policy class ((6), (7), (1), (2)) is a limiting special case of Gibbs soft-greedy optimistic policy iteration ((4), (5), (1), (2)).*

*Proof.* The update rule for Gibbs soft-greedy optimistic policy iteration is given in (4) and (5). By moving the temperature to scale $\hat{w}$ (assume $w_0$ to be scaled accordingly), we obtain

$$\begin{cases} w'_{k+1} & = w'_k + \kappa_k(\hat{w}_k/\tau_k - w'_k) \\ \theta_{k+1} & = w'_{k+1} \; , \end{cases} \tag{10}$$

again with $\kappa_k \in (0, 1]$, $\tau_k \in (0, \infty)$. Such a re-formulation effectively re-scales $w$ and is possible only with deep policy evaluation (cf. Section 2), with which the non-scaled $w$ is not needed by the policy evaluation process. We can now remove the redundant second line and rename $w'$ to $\theta$:

$$\theta_{k+1} = \theta_k + \kappa_k(\hat{w}_k/\tau_k - \theta_k) \; . \tag{11}$$

Finally, we open up the last term and encapsulate $\kappa/\tau$ into $\alpha$:

$$\theta_{k+1} = \theta_k + \kappa_k(\hat{w}_k/\tau_k) - \kappa_k \theta_k \tag{12}$$
$$= \theta_k + \alpha_k \hat{w}_k - \kappa_k \theta_k \; , \tag{13}$$

with $\alpha_k = \kappa_k/\tau_k$. Based on (7), we observe that (13) is equivalent to (8). The original natural actor-critic algorithm is obtained in the limit as $\kappa_k \to 0$, which causes the forgetting term $\kappa_k \theta_k$ to vanish (the effective step size $\alpha$ can still be controlled with $\tau$).

$\square$

This result has some interesting implications. First, it becomes apparent that the implicit effective step size in optimistic policy iteration is, in fact, $\alpha = \kappa/\tau$, i.e., it is inversely related to the temperature $\tau$. If the interpolation factor $\kappa$ is held fixed, a low temperature, which can lead to policy

oscillation, equals a long effective step size. This agrees with the interpretation of policy oscillation as overshooting in [24, 25]. Likewise, a high temperature equals a short effective step size. In [18], convergence is established for a high enough *constant* temperature. This result now becomes translated to showing that convergence is established with a short enough *constant* effective step size,[1] which creates an interesting and more direct connection to convergence results for (batch) steepest descent methods with a constant step size [e.g., 1, 3]. In addition, this connection *might* permit the application of the results in the aforementioned literature to establish, in the considered context, a constant step size convergence result for the natural actor-critic methodology.

Second, we see that the interpolation scheme in optimistic policy iteration, while originally introduced for the sake of countering an inaccurate value function estimate, actually goes in the direction of the policy gradient methodology. Smooth interpolation between policy gradient and greedy value function learning turns out to be possible by simply adjusting the interpolation factor $\kappa$ while treating the temperature $\tau$ as an inverse of the step size (we return to provide an interpretation of the role of $\kappa$ at a later point). Contrary to the related result in [24], no modifications to existing algorithms are needed. This connection also allows the convergence results from the policy gradient literature to be brought in (see Section 2): convergence w.p.1, under standard assumptions from the referred literature, to an optimal solution is established in the limit for this class of approximate optimistic policy iteration as the interpolation factor $\kappa$ is taken toward zero and the step size requirements are inversely enforced on the temperature $\tau$.

Third, we observe that in non-optimistic policy iteration ($\kappa = 1$), the forgetting term resets the parameter vector to the origin at the beginning of every iteration, with the implication that solutions that are not within the range of *a single step* from the origin in the direction of the natural gradient cannot be reached in any number of iterations. The choice of the effective step size, which is inversely controlled by the temperature, becomes again decisive: a step size that is too short (the temperature is too high) will cause the algorithm to permanently undershoot the desired optimum, thus trapping it in sustained sub-optimality, while a step size that is too long (the temperature is too low) will cause it to overshoot, which can additionally trap it in sustained oscillation. Unfortunately, even hitting the target exactly with a perfect step size will fail to lead to convergence and optimality at the same time. Our next section examines these issues more closely.

## 4 Systematic non-optimality of soft-greedy methods

For greedy value function methods, using the hard-greedy policy class trivially prevents convergence to other than deterministic policies. Furthermore, the proximity of an attractive stochastic policy can prevent convergence altogether and trap the process in oscillation (cf. Section 2). The Gibbs soft-greedy policy class, on the other hand, *can* represent stochastic policies, fixed points do exist [10, 17], and convergence toward *some* policy is guaranteed with sufficient softness [18, 14]. While convergence toward deterministic optimal decisions is trivially lost as soon as any softness is introduced ($\tau \nrightarrow 0$, and assuming a bounded value function), one might hope that convergence toward stochastic optimal decisions could still occur in some cases. Unfortunately, as we show in the following, this is not the case: in the presence of any softness, this approach can never converge toward any optimal policy (i.e., convergence and optimality become mutually exclusive), except in a certain pathological case.

At this point, we wish to make clear that we are not arguing against the *practical* value of the greedy value function methodology in (interactively) approximated problems; the methodology has some clear merits, and the sub-optimality and oscillations could well be negligible in a given task. Instead, we take the following result, together with existing literature on policy oscillations, as an indication of a fundamental theoretical incompatibility of this methodology to this context: the way by which this methodology deals with stochastic optima seems to be fundamentally flawed, and we believe that a thorough understanding of this flaw will have, in addition to facilitating sound theoretical advances, also immediate practical value by permitting correctly informed trade-off decisions.

**Theorem 2.** *Assume an unbiased value function estimator (e.g., Monte Carlo evaluation). Now, for Gibbs soft-greedy policy iteration ((1), (4) and (5)) using a linear-in-parameters value function approximator ((2) or (3)), including optimistic and non-optimistic variants (any $\kappa$ in (4)), there cannot exist a fixed point at an optimum, except for the uniformly stochastic policy.*

*Proof outline.* A fixed point of the update rule (4) must satisfy

$$\hat{w}_k = w_k \,, \tag{14}$$

i.e., at a fixed point, the policy evaluation step $\hat{w}_k := \mathrm{eval}(\pi(w_k/\tau_k))$ for the current parameter vector must yield the same parameter vector as its result:

$$\mathrm{eval}\left(\pi\left(w_k/\tau_k\right)\right) = w_k \,. \tag{15}$$

By applying (14) and (7), we have

$$w_k = \hat{w}_k = \eta_k = G(\theta_k)^{-1}\nabla_\theta J(\theta_k) \,, \tag{16}$$

which shows that the fixed-point policy $\pi(w_k/\tau_k)$ in (15) is defined solely by its own (scaled) performance gradient.

For an optimal policy and an unbiased estimator, this parameter gradient must, by definition, map to the zero policy gradient, i.e., to $\nabla_\pi J(\theta_k) = 0$. Consequently, an optimal policy at a fixed point is defined solely by the zero policy gradient, making the policy equal to $\pi(0)$, which is the uniformly stochastic policy. For the full proof, see the extended version.

**Theorem 3.** *Consider the family of methods from Theorem 2. Assume a smooth policy gradient field ($\|\nabla_\pi J(\pi_u) - \nabla_\pi J(\pi_v)\| \to 0$ as $\|\pi_u - \pi_v\| \to 0$) and $\tau \not\to 0$. First, the policy distance between a fixed point policy $\pi^f$ and an optimal policy $\pi^\star$ cannot be vanishingly small ($\|\pi^f - \pi^\star\| \not< \epsilon$), except if the optimal policy $\pi^\star$ is a semi-uniformly stochastic policy. Second, for bounded returns ($\gamma \not\to 1$ and $r(s,a) \not\to \pm\infty, \forall s, a$), the policy distance between a fixed point policy $\pi^f$ and an optimal policy $\pi^\star$ cannot be vanishingly small ($\|\pi^f - \pi^\star\| \not< \epsilon$), except if the optimal policy $\pi^\star$ is the uniformly stochastic policy.*

*Proof outline.* For a policy $\bar{\pi} = \pi(w_k/\tau_k)$ that is vanishingly close to an optimum, an unbiased parameter gradient $\eta_k$ must, assuming a smooth gradient field, map to a policy gradient that is vanishingly close to zero, i.e., $\eta_k$ must have a vanishingly small effect on $\bar{\pi}$ with any finite step size:

$$\|\pi(w_k/\tau_k + \alpha\eta_k) - \pi(w_k/\tau_k)\| < \epsilon \,, \quad \forall \alpha > 0, \ \alpha \not\to \infty \,. \tag{17}$$

If $\bar{\pi}$ is also a fixed point, then, by (16), we can substitute both $w_k$ and $\eta_k$ in (17) with $\hat{w}_k$:

$$\begin{aligned}
\|\pi(\hat{w}_k/\tau_k + \alpha\hat{w}_k) - \pi(\hat{w}_k/\tau_k)\| &< \epsilon \,, & \forall \alpha > 0, \ \alpha \not\to \infty \\
\Leftrightarrow \|\pi\left((1/\tau_k + \alpha)\hat{w}_k\right) - \pi((1/\tau_k)\hat{w}_k)\| &< \epsilon \,, & \forall \alpha > 0, \ \alpha \not\to \infty \,.
\end{aligned} \tag{18}$$

We now see that $\bar{\pi}$ is defined solely by a temperature-scaled version of a vanishingly small policy gradient, and that the condition in (17) is equivalent to stating that any finite decrease of the temperature must not have a non-vanishing effect on $\bar{\pi}$. As only semi-uniformly stochastic policies are invariant to such temperature decreases, it follows that $\bar{\pi}$ must be vanishingly close to such a policy.

Furthermore, if assuming bounded returns, then no dimension of the term $\hat{w}^\top \phi(s,a)$ can approach positive or negative infinity when $\hat{w}$ is estimated using (2) or (3). Consequently, for $\tau \not\to 0$, the uniformly stochastic policy $\pi(0)$ becomes the only semi-uniformly stochastic policy that the Gibbs policy class in (1) can approach, with the implication that $\bar{\pi}$ must be vanishingly close to the uniformly stochastic policy. For the full proof, see the extended version.

To interpret the preceding theorems, we observe that the gist of them is that, assuming a well-behaved gradient field, the closer the evaluated policy is to an optimum, the closer the target point of the next greedy update will be to the origin (in policy parameter space). At a fixed point, the policy parameter vector must equal the target point of the next update, causing convergence to or toward a policy that is exactly optimal but not at the origin to be a contradiction (Theorem 2). Convergence to or toward a policy that is vanishingly close to an optimum is also impossible, except if the optimum is (semi-)uniformly stochastic (Theorem 3).

In practical terms, Theorem 2 states that even if the task at hand and the chosen hyperparameters would allow convergence to some policy in a finite number of iterations, the resulting policy can

never contain optimal decisions, except for uniformly stochastic ones. Theorem 3 generalizes this result to the case of asymptotic convergence toward some limiting policy: for unbounded returns and any $\tau \not\to 0$, it is impossible to have asymptotic convergence toward any optimal decision in any state, except for semi-uniformly stochastic decisions, and for bounded returns and any $\tau \not\to 0$, it is impossible to have asymptotic convergence toward any non-uniform optimal decision in any state.

If convergence is to occur, then the limiting policy must reside "between" the origin and an optimum, i.e., the result must always undershoot the optimum that the learning process was influenced by. However, we can see in (15) that by decreasing the temperature $\tau$, it is possible to shift this point of convergence further away from the origin and closer to the optimum: in the limit of $\tau \to 0$, (15) can permit the parameter vector $\hat{w}$ to converge toward a point that approaches the origin while, at the same time, allowing the corresponding policy $\pi(\hat{w}/\tau)$ to converge toward a policy that is arbitrarily close to a distant optimum (one can also see that with $\tau \to 0$, the inequality in (18) becomes satisfied for any $\hat{w}_k$, due to $\alpha \not\to \infty$). Unfortunately, as we already know, such manipulation of the distance of the fixed point from an optimum by adjusting $\tau$ can ruin convergence altogether in non-Markovian problems. Perkins & Precup [18] report negative convergence results for non-optimistic iteration ($\kappa = 1$) with a too low $\tau$, while for optimistic iteration ($\kappa < 1$), Melo et al. [14] report a lack of positive results. Interestingly, this latter case is exactly what Theorem 1 addressed, showing that there actually *is* a way out and that it is by moving toward natural policy gradient iteration: decreasing the temperature $\tau$ toward zero causes the sub-optimality to vanish, while decreasing the interpolation factor $\kappa$ at the same rate prevents the effective step size from exploding.

Finally, we provide a brief discussion on some questions that may have occurred to the reader by now. First, how does the preceding fit with the well-known soundness of greedy value function methods in the Markovian case? The crucial difference between the Markovian case (fully observable and tabular) and the non-Markovian case (partially observable or non-tabular) follows from the standard result for MDPs that states that in the former, all optima must be deterministic (with the possibility of redundant stochastic optima) [e.g., 23, §A.2]. For the Gibbs policy class, deterministic policies reside at infinity in some direction in the parameter space, with two implications for the Markovian case. First, the distance to an optimum never decreases. Consequently, the value function, being a correction toward an optimum, never vanishes toward a 'neutral' state. Second, only the direction of an optimum is relevant, as the distance can be always assumed to be infinite. This implies that in, and only in Markovian problems, the value function never ceases to retain all necessary information about the current solution, while in non-Markovian problems, relying solely on the value function can lead to losing track of the current solution.

Second, when moving toward an optimum at infinity, how can the value function / natural gradient (encoded by $\hat{w} = \eta$) stay non-zero and continue to properly represent action values while the corresponding policy gradient $\nabla_\pi J(\theta)$ must approach zero at the same time? We note that the equivalence in (7) is between a value function and a *natural* gradient $\eta$. We then recall that the curvature of the Gibbs policy class turns into a plateau at infinity, onto which the policy becomes pushed when moving toward a deterministic optimum. The increasing discrepancy between $\eta = G(\theta)^{-1}\nabla_\theta J(\theta) \not\to 0$ and $\nabla_\pi J(\theta) \to 0$ can be consumed by $G(\theta)^{-1}$ as it captures the curvature of this plateau.

## 5   Common ground

Figure 1 shows a map of relevant variants of optimistic policy iteration, parameterized as in (4). As is well known, the hard-greedy variants of this methodology (seen on the left edge on the map) can become trapped in non-converging cycles over potentially non-optimal policies (see Section 2 for references and exceptions). For a continuously soft-greedy policy class (toward right on the map), convergence can be established with enough softness [18, 14]. The natural actor-critic algorithm, which is convergent and optimal, is placed to the lower left corner by Theorem 1, while the inevitable non-optimality of soft-greedy variants toward right follows from Theorems 2 and 3. The exact (problem-dependent) place and shape of the line separating non-convergent and convergent soft-greedy variants (dashed line on the map) remains an open problem.

The main value of Theorem 1 is in bringing the greedy value function and policy gradient methodologies closer to each other. In our context, the unifying NAC($\kappa$) formulation in (8) permits interpolation between the methodologies using the $\kappa$ parameter. As discussed at the end of Section 4, the policy-forgetting term requires a Markovian problem for being justified: a greedy update implicitly

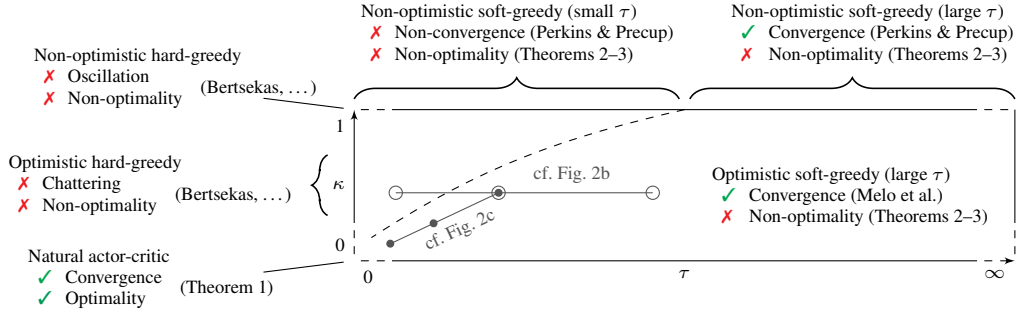

Figure 1: The hyperparameter space of the general form of (approximate) optimistic policy iteration in (4), with known convergence and optimality properties (see text for assumptions).

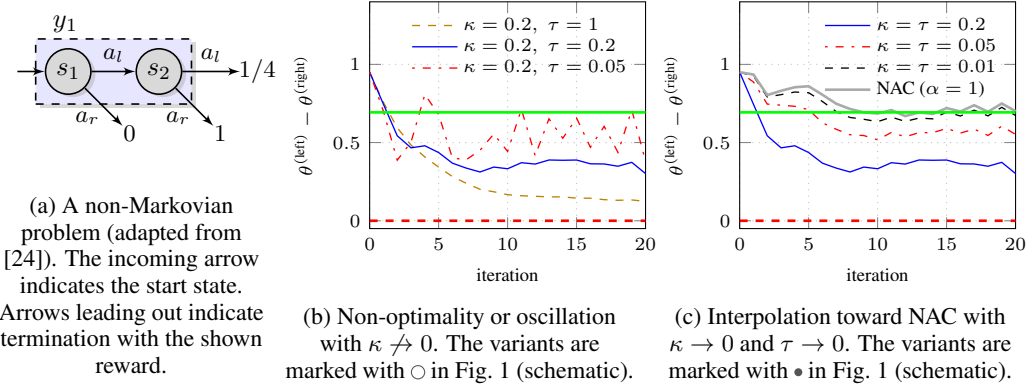

(a) A non-Markovian problem (adapted from [24]). The incoming arrow indicates the start state. Arrows leading out indicate termination with the shown reward.

(b) Non-optimality or oscillation with $\kappa \not\to 0$. The variants are marked with $\circ$ in Fig. 1 (schematic).

(c) Interpolation toward NAC with $\kappa \to 0$ and $\tau \to 0$. The variants are marked with $\bullet$ in Fig. 1 (schematic).

Figure 2: Empirical illustration of the behavior of optimistic policy iteration ((1), (2), (4) and (5), with tabular $\phi$) in the proximity of a stochastic optimum. The problem is shown in Fig. 2a. In Figures 2b and 2c, the optimum at $\theta^{(\text{left})} - \theta^{(\text{right})} = \log(2)$ is denoted by a solid green line. The uniformly stochastic policy is denoted by a dashed red line.

stands on a Markov assumption and the $\kappa$ parameter in (8) can be interpreted as adjusting the strength of this assumption. In this respect, the policy improvement parameter $\kappa$ in NAC($\kappa$) can be seen (inversely) as a dual in spirit to the policy evaluation parameter $\lambda$ in TD($\lambda$)-style algorithms. On the policy evaluation side, having $\lambda = 0$ obtains variance reduction by assuming and exploiting Markovianity of the problem, while $\lambda = 1$ obtains unbiased estimates also for non-Markovian problems. On the policy improvement side, with $\kappa = 1$, we have strictly greedy updates that gain in speed as the policy can respond instantly to new opportunities appearing in the value function (for empirical observations of such a speed gain, see [11, 25]), and in representational flexibility due to the lack of continuity constraints between successive policies (for a canonical example, consider fitted Q iteration). This comes at the price of either oscillation or non-optimality if the Markov assumption fails to hold, which is illustrated in Figure 2b for the problem in 2a. With $\kappa \to 0$, we approach natural gradient updates that remain sound also in non-Markovian settings, which is illustrated in Figure 2c. The possibility to interpolate between the approaches might turn out useful in problems with partial Markovianity: a large $\kappa$ in the NAC($\kappa$) formulation can be used to quickly find the rough direction of the strongest attractors, after which gradually decreasing $\kappa$ allows a convergent final ascent toward an optimum.

## Acknowledgments

This work has been financially supported by the Academy of Finland through project no. 254104, and by the Foundation of Nokia Corporation.

## Footnotes

[1]Note that the diminishing step size $\alpha_t$ in [18, Fig. 1] concerns policy *evaluation*, not policy *improvement*.

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
