[Supplementary Material]

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

 detailed description of the oscillation phenomenon can be found in [7, §6.4; 6, §3.5], where it is described in terms of cyclic sequences on the so-called greedy partition of the value function parameter space. It is also noted that the properties of the policy evaluation method (including properties that affect evaluation depth) do not seem to affect the ultimate cycles of policies. It is possible to obtain asymptotic parameter convergence by employing an optimistic policy iteration scheme, but even then the corresponding policy can continue oscillating. This is known as policy chattering. In such cases, the value function parameters spiral toward an attractor on a boundary of the greedy partition, and the obtained value function can fail to meaningfully represent the expected returns of any of the involved policies [6, §3.5]. A novel explanation to the phenomenon in the non-optimistic case was recently proposed in [26, 27], where policy oscillation was re-cast as sustained overshooting over an attractive stochastic policy.

Policy convergence can be established under various restrictions. A common approach is to use $\theta$-independent sampling (non-interactivity), which enables convergence for a well-studied family of approximators (for reviews of this methodology, see [28, §2] and [9]). Interestingly, with the

$\theta$-independent aggregation approach introduced in [6, 5], convergence is obtained without completely sacrificing interactivity. However, the approach is based on a $\theta$-independent projection mapping, which effectively causes representational resources to be allocated in a fixed manner instead of according to the on-policy distribution; one of the natural benefits of interactivity becomes lost, although whether this has practical implications remains to be seen. In the case of Monte Carlo estimation of action values, it is also possible to establish convergence by solely modifying the exploration scheme [e.g., 10]. Importantly to this paper, convergence can be established also with continuously soft-greedy action selection [20, 16], in which case, however, the quality of the obtained solutions is unknown. Finally, the analysis in [12] bears some resemblance to our first main result. The key to seeing the relation is in noting that their main update rule (their Eq. (4.1)) is equivalent to our Eq. (4), except for being defined in the *policy* space. While they establish convergence up to a "breaking point" under relatively restrictive assumptions, we proceed by directly connecting to stronger and more generic results from the policy gradient literature.

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

**Lemma 1.** *Consider the Gibbs/Boltzmann distribution* $\mathbb{P}(i|\omega) = e^{\omega^\top \mathbb{I}(i)} / \sum_z e^{\omega^\top \mathbb{I}(z)}$, *where* $\omega \in \mathbb{R}^n$, *and* $\mathbb{I}(i) \in \mathbb{R}^n$ *is the indicator function that picks the $i$th element of its multiplier. This distribution is invariant to, and only to, uniform translations of its argument vector $\omega$:*

$$\frac{e^{(\omega+c)^\top \mathbb{I}(i)}}{\sum_z e^{(\omega+c)^\top \mathbb{I}(z)}} = \frac{e^{\omega^\top \mathbb{I}(i)}}{\sum_z e^{\omega^\top \mathbb{I}(z)}} , \quad \forall i , \tag{14}$$

*where* $c \in \mathbb{R}^n$, *if and only if*

$$c^\top \mathbb{I}(i) = c^\top \mathbb{I}(z) , \quad \forall i, z . \tag{15}$$

*Furthermore, the Gibbs policy class in* (1) *is invariant to, and only to, per-state uniform translations of the term* $\theta^\top \phi(s, a)$, *and if some translation of the parameter vector $\theta$ operates within this invariance for some $\theta_x$, then it does so for all $\theta$.*

*Proof.* Consider some translation $c$ of the argument vector $\omega$ of the Gibbs/Boltzmann distribution $\mathbb{P}(i|\omega) = e^{\omega^\top \mathbb{I}(i)} / \sum_z e^{\omega^\top \mathbb{I}(z)}$, with $\omega, c \in \mathbb{R}^n$, and $\mathbb{I}(i) \in \mathbb{R}^n$ as the indicator function that picks the $i$th element of its multiplier:

$$\frac{e^{(\omega+c)^\top \mathbb{I}(i)}}{\sum_z e^{(\omega+c)^\top \mathbb{I}(z)}} = \frac{e^{\omega^\top \mathbb{I}(i)+c^\top \mathbb{I}(i)}}{\sum_z e^{\omega^\top \mathbb{I}(z)+c^\top \mathbb{I}(z)}} = \frac{e^{\omega^\top \mathbb{I}(i)} e^{c^\top \mathbb{I}(i)}}{\sum_z e^{\omega^\top \mathbb{I}(z)} e^{c^\top \mathbb{I}(z)}} = \frac{e^{\omega^\top \mathbb{I}(i)}}{\sum_z e^{\omega^\top \mathbb{I}(z)}} , \quad \forall i , \tag{16}$$

if and only if

$$c^\top \mathbb{I}(i) = c^\top \mathbb{I}(z) , \quad \forall i, z . \tag{17}$$

Importantly to us, whether translating by some $c$ affects the distribution or not, does not depend on the value of $\omega$, as $\omega$ does not appear in (17): if a given $c$ is neutral for some $\omega$, then it is neutral for all $\omega$.

We note that the condition (17) only requires $c^\top \mathbb{I}(i)$ to evaluate to the same value for all $i$, but this condition does not take a stance on what happens inside this term. Thus, we can replace, without modifying the form of the requirement, the indicator function $\mathbb{I}(i)$ with a function $f(\cdot) \in \mathbb{R}^n$ that picks an arbitrary linear combination of elements from its multiplier:

$$c^\top f(i) = c^\top f(z) , \quad \forall i, z . \tag{18}$$

Finally, we can extend $f$ to accept an additional argument $k$, with the consequence that (18) must hold simultaneously for all $k$ to permit the last step in (16):

$$\frac{e^{(\omega+c)^\top f(k,i)}}{\sum_z e^{(\omega+c)^\top f(k,z)}} = \frac{e^{\omega^\top f(k,i)}}{\sum_z e^{\omega^\top f(k,z)}} , \quad \forall k, i \tag{19}$$

$$\Leftrightarrow c^\top f(k,i) = c^\top f(k,z) , \quad \forall k, i, z . \tag{20}$$

(19) and (20) now apply to (1), stating that (1) is invariant to, and only to, per-state uniform translations of the term $\theta^\top \phi(s, a)$, and that if some translation $c$ of the parameter vector $\theta$ operates within this invariance for some $\theta_x$, then it does so for all $\theta$:

$$
\begin{aligned}
&\pi(a|s, \theta_x + c) = \pi(a|s, \theta_x) \quad \forall s, a \\
\Leftrightarrow\ &\pi(a|s, \theta + c) = \pi(a|s, \theta) \quad \forall s, a, \theta \ .
\end{aligned}
\tag{21}
$$

$\square$

**Theorem 2.** *Assume an unbiased value function estimator (e.g., Monte Carlo evaluation). Now, for Gibbs soft-greedy policy iteration ((1), (4) and (5)) using a linear-in-parameters value function approximator ((2) or (3)), including optimistic and non-optimistic variants (any $\kappa$ in (4)), there cannot exist a fixed point at an optimum, except for the uniformly stochastic policy.*

*Proof outline.* A fixed point of the update rule (4) must satisfy

$$
\hat{w}_k = w_k \ ,
\tag{22}
$$

i.e., at a fixed point, the policy evaluation step $\hat{w}_k := \text{eval}(\pi(w_k/\tau_k))$ for the current parameter vector must yield the same parameter vector as its result:

$$
\text{eval}\left(\pi\left(w_k/\tau_k\right)\right) = w_k \ .
\tag{23}
$$

By applying (22) and (7), we have

$$
w_k = \hat{w}_k = \eta_k = G(\theta_k)^{-1} \nabla_\theta J(\theta_k) \ ,
\tag{24}
$$

which shows that the fixed-point policy $\pi(w_k/\tau_k)$ in (23) is defined solely by its own (scaled) performance gradient.

For an optimal policy and an unbiased estimator, this parameter gradient must, by definition, map to the zero policy gradient, i.e., to $\nabla_\pi J(\theta_k) = 0$. Consequently, an optimal policy at a fixed point is defined solely by the zero policy gradient, making the policy equal to $\pi(0)$, which is the uniformly stochastic policy.

*Proof.* By definition, when the policy defined by some $\theta_x \in \Theta$ is locally optimal, the parameter gradient vector $\eta_{x,0} = G(\theta_x)^{-1} \nabla_\theta J(\theta_x)$ estimated by an unbiased estimator must be such that it maps to the zero policy gradient $\nabla_\pi J(\theta_x) = 0$ at $\theta_x$, i.e., it does not change $\pi$ at $\theta_x$:

$$
\pi(\theta_x + \alpha \eta_{x,0}) = \pi(\theta_x) \ , \quad \forall \alpha \ ,
\tag{25}
$$

with $\pi(\cdot)$ from (1). For this to be possible, $\eta_{x,0}$ must fulfill (20) (in the role of $c$), which allows us to apply (21) and extend (25) from $\theta_x$ to all $\theta$:

$$
\begin{aligned}
&\pi(\theta_x + \alpha \eta_{x,0}) = \pi(\theta_x) \ , \quad \forall \alpha \\
\Leftrightarrow\ &\pi(\theta + \alpha \eta_{x,0}) = \pi(\theta) \ , \quad \forall \alpha, \theta \ .
\end{aligned}
\tag{26}
$$

Most importantly, (26) implies the following:

$$
\begin{aligned}
&\pi(\theta_x + \alpha \eta_{x,0}) = \pi(\theta_x) \ , \quad \forall \alpha \\
\Leftrightarrow\ &\pi(\alpha \eta_{x,0}) = \pi(0 + \alpha \eta_{x,0}) = \pi(0) \ , \quad \forall \alpha \ ,
\end{aligned}
\tag{27}
$$

where $\pi(0)$ is the uniformly stochastic policy. In words, whenever the evaluated policy is locally optimal, an unbiased gradient estimator must produce such a parameter gradient vector $\eta_{x,0}$ that when added to any policy parameter vector $\theta$, *including the zero parameter vector*, does not change the resulting policy.

A fixed point of the update rule (4) must satisfy

$$
\hat{w}_k = w_k \ ,
\tag{28}
$$

regardless of the value of $\kappa$. By applying (5), we see that the limiting policy at a fixed point is

$$
\pi(\hat{w}_k/\tau_k) \ .
\tag{29}
$$

For the case of (2), we can apply (7) and re-write this policy as

$$\pi(\alpha_k \eta_k) \,, \tag{30}$$

with $\alpha_k = 1/\tau_k$. Also, the combination of (1) and (2) forms a 'compatible' actor-critic setup, implying that with an unbiased value function estimator, $\eta_k$ is an unbiased gradient estimate (Section 2). Consequently, for a locally optimal policy, $\eta_k$ must fulfill (27), with the implication that (30) is the uniformly stochastic policy. Thus, under the given assumptions, whenever the policy is simultaneously locally optimal and a fixed point of (4), it must be the uniformly stochastic policy.

Extension from (2) to (3) follows by noting that the definitions of $\bar{Q}$ and $Q$ differ only by a per-state uniform translation: $Q(s,a) = \bar{Q}(s,a) + V(s)$. As observed in [24], such a translation does not break the compatibility condition, i.e., it does not affect the biasedness of the corresponding gradient. This can be seen also by noting that the translation in $Q(s,a) = \hat{w}^\top \phi(s,a)$ is carried, via (29), to a per-state uniform translation of the term $\theta^\top \phi(s,a)$ in (1), to which the policy class was shown in (19) and (20) to be invariant. Consequently, $\hat{w}_k/\tau_k$ in (29) still defines an unbiased gradient estimate and, for a locally optimal policy, (27) stays in effect.

$\square$

This result still leaves open the possibility of a fixed point residing vanishingly close to an optimum, in which case asymptotic convergence toward such a fixed point would mean asymptotic convergence toward the adjacent optimum. This possibility is ruled out with the following theorem that shows, assuming a smooth gradient field and $\tau \not\to 0$, that the distance between a fixed point and an optimum cannot be vanishingly small, with the exception of an optimum that is semi-uniformly stochastic (assuming unbounded returns) or uniformly stochastic (assuming bounded returns).

**Lemma 2.** *For the Gibbs/Boltzmann distribution $\mathbb{P}(i|\omega) = e^{(\omega/\tau)^\top \mathbb{I}(i)} / \sum_z e^{(\omega/\tau)^\top \mathbb{I}(z)}$, any finite decrease of the temperature $\tau$ has a vanishingly small effect on the distribution if and only if the initial distribution is vanishingly close to being semi-uniformly stochastic (we simplify our notation by noting that decreasing the temperature is equivalent to upscaling the parameter vector):*

$$\left| \frac{e^{\omega^\top \mathbb{I}(i)}}{\sum_z e^{\omega^\top \mathbb{I}(z)}} - \frac{e^{(\alpha\omega)^\top \mathbb{I}(i)}}{\sum_z e^{(\alpha\omega)^\top \mathbb{I}(z)}} \right| < \epsilon \,, \qquad \forall i, \ \forall \alpha > 1, \ \alpha \not\to \infty \tag{31}$$

$$\Leftrightarrow \left| \frac{e^{\omega^\top \mathbb{I}(i)}}{\sum_z e^{\omega^\top \mathbb{I}(z)}} - c \right| < \epsilon \quad \vee \quad \frac{e^{\omega^\top \mathbb{I}(i)}}{\sum_z e^{\omega^\top \mathbb{I}(z)}} < \epsilon \,, \qquad \forall i, \ \exists c \in [0,1] \,. \tag{32}$$

*Proof.* Informally, this holds because decreasing the temperature makes the distribution more deterministic by amplifying the probability differences present. For this to have no non-vanishing effect on the distribution for any finite decrease of the temperature, it must be that there are no non-vanishing probability differences in non-vanishingly weak dimensions to be amplified, i.e., for all dimensions for which the distribution is not close to zero it must be close to some common constant.

Consider a parameter vector $\bar{\omega}$ that defines some initial Gibbs distribution. Let us choose one of the maximizing dimensions of this distribution and denote it by $i^\star$:

$$\frac{e^{\bar{\omega}^\top \mathbb{I}(i^\star)}}{\sum_z e^{\bar{\omega}^\top \mathbb{I}(z)}} \geq \frac{e^{\bar{\omega}^\top \mathbb{I}(j)}}{\sum_z e^{\bar{\omega}^\top \mathbb{I}(z)}} \,, \quad \forall j \,. \tag{33}$$

There might also exist some dimensions by which the distribution is non-vanishingly smaller than for the maximizing dimension $i^\star$, in case that the distribution is not close to being uniformly stochastic. Let us denote the (possibly empty) set of these dimensions by $K$:

$$K = \left\{ k \left| \frac{e^{\bar{\omega}^\top \mathbb{I}(i^\star)}}{\sum_z e^{\bar{\omega}^\top \mathbb{I}(z)}} \gg \frac{e^{\bar{\omega}^\top \mathbb{I}(k)}}{\sum_z e^{\bar{\omega}^\top \mathbb{I}(z)}} \right. \right\} \,, \tag{34}$$

where $\gg$ is a shorthand for a non-vanishing inequality. We then de-normalize (33) and (34) into

$$\begin{cases} e^{\bar{\omega}^\top \mathbb{I}(i^\star)} \geq e^{\bar{\omega}^\top \mathbb{I}(j)} \,, & \forall j \,, \tag{35a} \\ e^{\bar{\omega}^\top \mathbb{I}(i^\star)} \gg e^{\bar{\omega}^\top \mathbb{I}(k)} \,, & \forall k \in K \,. \tag{35b} \end{cases}$$

We can see that for the near-equality in (31) to hold for the considered $\bar{\omega}$ and for the $i^\star$ chosen, i.e.,

$$\left| \frac{e^{\bar{\omega}^\top \mathbb{I}(i^\star)}}{\sum_z e^{\bar{\omega}^\top \mathbb{I}(z)}} - \frac{e^{(\alpha\bar{\omega})^\top \mathbb{I}(i^\star)}}{\sum_z e^{(\alpha\bar{\omega})^\top \mathbb{I}(z)}} \right| < \epsilon , \quad \forall \alpha > 1, \ \alpha \not\to \infty , \tag{36}$$

it must be that $\alpha$ makes both the numerator and the denominator of the affected fraction to grow (or shrink for a negative $\bar{\omega}^\top \mathbb{I}(i^\star)$) nearly exactly as fast, so that the effect of $\alpha$ nearly cancels out. Let us use $\sigma_i$ to denote the scaling factor by which $\alpha$ scales the terms $e^{(\alpha\bar{\omega})^\top \mathbb{I}(i)}$:

$$\sigma_i = \frac{e^{(\alpha\bar{\omega})^\top \mathbb{I}(i)}}{e^{\bar{\omega}^\top \mathbb{I}(i)}} . \tag{37}$$

We note that $\alpha$ affects the value of these terms via linear operations and an exponentiation with a positive base, making the effect of $\alpha$ strictly monotone. By combining this monotonicity with (35a), it follows that $\alpha$ can make no term of the affected denominator sum in (36) to grow faster (or shrink slower) than the affected numerator, and, by furthermore combining this monotonicity with (35b), it follows that $\alpha$ will make all terms of the affected denominator sum in (36) that correspond to the dimensions in $K$ to grow non-vanishingly slower (or to shrink non-vanishingly faster for a negative $\bar{\omega}^\top \mathbb{I}(i^\star)$) than the affected numerator:

$$\begin{cases} \sigma_{i^\star} \geq \sigma_j , & \forall j \\ \sigma_{i^\star} \gg \sigma_j , & \forall j \in K . \end{cases} \tag{38a} \tag{38b}$$

We now see from (38) that for the affected fraction in (36) to stay nearly constant, it must be that *all* of the terms in the denominator sum that correspond to the dimensions in $K$ have a vanishingly small effect on the sum: if any non-vanishingly contributing term is affected by (38b), there can be no other term that would compensate for this change in the sum, due to (38a). In other words, for the near-equality in (31) to hold for the considered $\bar{\omega}$ and for the maximizing $i^\star$ chosen earlier, every term in the denominator sum $\sum_z e^{\bar{\omega}^\top \mathbb{I}(z)}$ of the initial distribution must be either a) vanishingly close to the term for $i^\star$ ($e^{\bar{\omega}^\top \mathbb{I}(i^\star)} \not\gg e^{\bar{\omega}^\top \mathbb{I}(z)}$, so that $z \notin K$), or b) have a vanishingly small effect on the sum ($e^{\bar{\omega}^\top \mathbb{I}(z)} / \sum_{z'} e^{\bar{\omega}^\top \mathbb{I}(z')} < \epsilon$):

$$e^{\bar{\omega}^\top \mathbb{I}(i^\star)} - e^{\bar{\omega}^\top \mathbb{I}(z)} < \epsilon \quad \vee \quad \frac{e^{\bar{\omega}^\top \mathbb{I}(z)}}{\sum_{z'} e^{\bar{\omega}^\top \mathbb{I}(z')}} < \epsilon , \quad \forall z . \tag{39}$$

By comparing to the semi-uniformity requirement in (32) while letting $c = e^{\bar{\omega}^\top \mathbb{I}(i^\star)}$, it can be seen that (39) now defines a policy that is vanishingly close to a semi-uniformly stochastic policy. Extension to the Gibbs policy in (1) follows directly by replacing $\mathbb{I}(i)$ with $f(k,i)$ as in Lemma 1.

$\square$

**Theorem 3.** *Consider the family of methods from Theorem 2. Assume a smooth policy gradient field ($\|\nabla_\pi J(\pi_u) - \nabla_\pi J(\pi_v)\| \to 0$ as $\|\pi_u - \pi_v\| \to 0$) and $\tau \not\to 0$. First, the policy distance between a fixed point policy $\pi^f$ and an optimal policy $\pi^\star$ cannot be vanishingly small ($\|\pi^f - \pi^\star\| \not< \epsilon$), except if the optimal policy $\pi^\star$ is a semi-uniformly stochastic policy. Second, for bounded returns ($\gamma \not\to 1$ and $r(s,a) \not\to \pm\infty, \forall s,a$), the policy distance between a fixed point policy $\pi^f$ and an optimal policy $\pi^\star$ cannot be vanishingly small ($\|\pi^f - \pi^\star\| \not< \epsilon$), except if the optimal policy $\pi^\star$ is the uniformly stochastic policy.*

*Proof outline.* For a policy $\bar{\pi} = \pi(w_k/\tau_k)$ that is vanishingly close to an optimum, an unbiased parameter gradient $\eta_k$ must, assuming a smooth gradient field, map to a policy gradient that is vanishingly close to zero, i.e., $\eta_k$ must have a vanishingly small effect on $\bar{\pi}$ with any finite step size:

$$\|\pi(w_k/\tau_k + \alpha\eta_k) - \pi(w_k/\tau_k)\| < \epsilon , \quad \forall \alpha > 0, \ \alpha \not\to \infty . \tag{40}$$

If $\bar{\pi}$ is also a fixed point, then, by (24), we can substitute both $w_k$ and $\eta_k$ in (40) with $\hat{w}_k$:

$$\begin{aligned} &\|\pi(\hat{w}_k/\tau_k + \alpha\hat{w}_k) - \pi(\hat{w}_k/\tau_k)\| < \epsilon , & \forall \alpha > 0, \ \alpha \not\to \infty \\ \Leftrightarrow &\|\pi((1/\tau_k + \alpha)\hat{w}_k) - \pi((1/\tau_k)\hat{w}_k)\| < \epsilon , & \forall \alpha > 0, \ \alpha \not\to \infty . \end{aligned} \tag{41}$$

We now see that $\bar{\pi}$ is defined solely by a temperature-scaled version of a vanishingly small policy gradient, and that the condition in (40) is equivalent to stating that any finite decrease of the temperature must not have a non-vanishing effect on $\bar{\pi}$. As only semi-uniformly stochastic policies are invariant to such temperature decreases, it follows that $\bar{\pi}$ must be vanishingly close to such a policy.

Furthermore, if assuming bounded returns, then no dimension of the term $\hat{w}^\top \phi(s, a)$ can approach positive or negative infinity when $\hat{w}$ is estimated using (2) or (3). Consequently, for $\tau \not\to 0$, the uniformly stochastic policy $\pi(0)$ becomes the only semi-uniformly stochastic policy that the Gibbs policy class in (1) can approach, with the implication that $\bar{\pi}$ must be vanishingly close to the uniformly stochastic policy.

*Proof.* Consider some policy $\bar{\pi} = \pi(w_k/\tau_k)$ from the Gibbs policy class (1) that is vanishingly close to a local optimum $\pi^\star$:

$$\|\pi(w_k/\tau_k) - \pi^\star\| < \epsilon . \tag{42}$$

By definition, for a smooth policy gradient field, the parameter gradient vector $\eta_k = G(\bar{\pi})^{-1} \nabla_\theta J(\bar{\pi})$ estimated for this policy by an unbiased value function estimator and approximator (e.g., Monte Carlo evaluation with (2)) must be such that it maps to a vanishingly small policy gradient $\|\nabla_\pi J(\bar{\pi})\| < \epsilon$, i.e., $\eta_k$ has a vanishingly small effect on $\bar{\pi}$ as long as the step size is finite:

$$\|\pi(w_k/\tau_k + \alpha\eta_k) - \pi(w_k/\tau_k)\| < \epsilon , \quad \forall \alpha > 0, \ \alpha \not\to \infty . \tag{43}$$

Now assume that $\bar{\pi}$ is a limiting fixed point policy of the update rule (4) when using the policy mapping (5). It follows immediately from (4) that such a fixed point must satisfy

$$\hat{w}_k = w_k , \tag{44}$$

regardless of the value of $\kappa$. If the policy evaluation step $\hat{w}_k := \mathrm{eval}(\pi(w_k/\tau_k))$ is performed using (2), we can directly use (7), which leads to

$$\hat{w}_k = \eta_k . \tag{45}$$

Also, the combination of (1) and (2) forms a 'compatible' actor-critic setup, implying that with an unbiased value function estimator, $\hat{w}_k = \eta_k$ is an unbiased gradient estimate (Section 2). Extension from (2) to (3) follows as in Theorem 2, with the implication that $\hat{w}_k$ is an unbiased gradient estimate also when using (3). This now allows us to take (43) and apply (44) and (45), so as to substitute both $w_k$ and $\eta_k$ with $\hat{w}_k$:

$$\begin{aligned} &\|\pi(\hat{w}_k/\tau_k + \alpha\hat{w}_k) - \pi(\hat{w}_k/\tau_k)\| < \epsilon , && \forall \alpha > 0, \ \alpha \not\to \infty \\ \Leftrightarrow \ &\|\pi\left((1/\tau_k + \alpha)\hat{w}_k\right) - \pi((1/\tau_k)\hat{w}_k)\| < \epsilon , && \forall \alpha > 0, \ \alpha \not\to \infty . \end{aligned} \tag{46}$$

We observe that the sole effect of $\alpha$ in (46) is that it decreases the temperature of the policy, i.e., it makes the policy more deterministic by amplifying the probability differences present in $\pi(\hat{w}_k/\tau_k) = \pi(w_k/\tau_k) = \bar{\pi}$. At the same time, the inequality requirement in (46) states that such amplification must have a vanishingly small effect on the policy *for any finite* $\alpha$: even huge changes in the policy temperature must have a vanishingly small effect on the policy. As shown by Lemma 2, this is possible only if $\bar{\pi}$ is vanishingly close to a semi-uniformly stochastic policy, so that there are no non-vanishing probability differences in non-vanishingly weak dimensions to be amplified:

$$|\bar{\pi}(a|s) - c_s| < \epsilon \quad \vee \quad \bar{\pi}(a|s) < \epsilon , \qquad \forall s, a, \ \forall s \ \exists c_s \in [0, 1] . \tag{47}$$

The latter case ($\bar{\pi}(a|s) < \epsilon$) is possible for the considered Gibbs policy class only if the term $\theta_k^\top \phi(s, a)$ in (1) can approach negative infinity or some other term $\theta_k^\top \phi(s, z)$ can approach positive infinity (assuming a finite action space). In our case, this translates to the following requirement:

$$(\hat{w}_k/\tau_k)^\top \phi(s, a) \to \pm\infty , \quad \exists s, a . \tag{48}$$

We recall that with both (2) and (3), $\hat{w}_k^\top \phi(s, a)$ (without the temperature parameter) is a least-squares fit to the estimated value function, possibly with a baseline shift toward centered values when using (2). Consequently, $\hat{w}_k^\top \phi(s, a)$ can approach infinity only if the estimated value function can approach infinity, which, for an unbiased value function estimator, is possible only if the returns

$\sum_t \gamma^t \mathbb{E}[r(S_t, A_t)]$ can approach infinity. This requires that either $\gamma \to 1$ or $r(s,a) \to \pm\infty$, $\exists s, a$. This applies also to $(\hat{w}_k/\tau_k)^\top \phi(s,a)$ (with the temperature parameter) as long as $\tau \nrightarrow 0$.[2] Thus, we have for $\tau \nrightarrow 0$ that (48) and consequently the latter case in (47) can hold only if

$$\gamma \to 1 \quad \vee \quad r(s,a) \to \pm\infty \ , \ \exists s, a \ . \tag{49}$$

In summary, a fixed point policy $\bar{\pi}$ that is vanishingly close to a local optimum must fulfill (47), i.e., it must be vanishingly close to a semi-uniformly stochastic policy. Furthermore, if the conditions in (49) are not allowed, then $\

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

[2]With $\tau_k \to 0$, the sum $1/\tau_k + \alpha$ in (46) becomes dominated by the first term (due to $\alpha \nrightarrow \infty$) and the inequality becomes satisfied for any $\hat{w}_k$. We return to this in the following discussion.