[Reviews · NeurIPS 2013]

Submitted by Assigned_Reviewer_6

The paper unifies the optimistic policy Iteration and the natural actor critic (NAC) approach for Gibbs policies. It presents a slight modification of the NAC algorithm, where the original algorithm is a special case which is called forgetful NAC. The authors show that forget full Nac and optimistic policy iteration are equivalent. The authors also present a non-optimality result for soft-greedy Gibbs distribution, I.e., the optimal solution is not a fixed point of the policy iteration algorithm.

I liked the unified view on both type of algorithms. While the new presented algorithm is equivalent to the existing optimistic policy iteration, the relationship between the natural actor critic and the value-based optimistic policy iteration algorithm where very interesting to me. I think the paper might be an important step to get a better understanding of both methods. The main weakness of the paper are the experiments where I think it would be good to show the effect of surpressing oscillations also for more complex tasks.

Clarity: The paper is well written. Some parts should be clarified. for example the definition of the action value function in line 100 (where one function is actually an advantage function). The intuition of optimistic policy iteration (line 122) could also be better explained.

Relevance: the paper is relevant for the RL community to get a better understanding of the NAC methods.

Significance: the paper presents an important step to get a better understanding of the NAC methods.

Minor issues: - what do you mean with w.p.l. ? I could not even find this abbreviation in Google...
- line 99: the parameter vectors w_k are not properly introduced
Summary: The paper presents a nice unified view on natural actor critic and optimistic policy iteration algorithms. The weakness of the paper are the experiments, which only cover a toy task.

Submitted by Assigned_Reviewer_7

This paper is interesting. The author has developed some new ways of thinking about existing reinforcement-learning algorithms and the relationships between them. A new "forgetful" actor-critic algorithm is proposed which brings policy-gradient and action-value methods into closer alignment; it can be proved to be equivalent to a softmax action-value method in a certain sense. It is hoped that these understandings might enable better convergence results or new algorithms with better convergence properties. However, it seems fair to me to say that not much has come out of this yet; it is mostly potential. And the work is very technical and detailed. We might ask the authors to keep going, and wish them luck, but not want to see the details of their special ways of viewing these algorithms until after they have borne fruit. I guess this is my recommendation. Right now these inconclusive improvements in understanding will likely be interesting to very few NIPS readers. The paper might be a good one for a more specialized meeting.


---
After feedback and seeing the other reviews:

I am glad to see that other reviewers also say the interesting points of this paper. Maybe it is not such a specialized topic after all. I will raise my assessment a notch.

Summary: Some deep thinking about the detailed relationships between major classes of reinforcement learning algorithms, but which has not yet reached a conclusion that is meaningful to anyone outside a tiny sliver of the reinforcement-learning research community. Might be suitable for a more specialized meeting than NIPS.

Submitted by Assigned_Reviewer_8

The paper "Optimistic policy iteration and natural actor-critic: A unifying view and a non-optimality result" investigates policy iteration and natural actor critic (NAC) methods for approximate solutions of Markov decision processes. The setting considers Boltzmann policies, compatible function approximation, and unbiased estimators. The authors introduce a forgetting factor in the NAC update (parameterized by a weight kappa), and show that this version is identical to an incremental ("optimistic") policy iteration (OPI) with learning rate kappa. By making the weight kappa go to zero and decreasing at the same time the policy temperature, the two algorithms tend towards a common limit corresponding to the original NAC (with policies that tend to fully greedy ones). This is exploited to derive insight on the two algorithms and how the convergence of OPI might be established using NAC; as well as to interpret the policy oscillation phenomenon in policy iteration. Then, the authors show that when the temperature is not allowed to decrease to 0, the unified algorithm cannot converge to any nontrivial optimal solution, for any learning rate. Together, the results suggest a good OPI algorithm might be obtained by allowing the parameters to tend to the limits discussed above (both weight kappa and the temperature going to zero, with the overall algorithm thus tending towards NAC). This is illustrated with simulations in a toy problem.

I think this is an insightful paper and has the potential to make a significant impact in the field of approximate reinforcement learning and dynamic programming. The way the authors look at the problem is original. The survey of existing convergence results is very nice.

While the paper is well-written in general, the detailed technical discussions are often rather dense and not easy to read; therefore, some work to alleviate this would be well spent. In particular, getting the intuition behind the discussion on the relation between standard NAC and the OPI limiting case (e.g. the next-to-last paragraph of Section 3) took a bit of rereading in my case.

I also have some non-major comments on the formalism.
- The authors should be careful to introduce all notations properly. E.g. J may not be explained in the first paragraph of Section 2; the Qbar notation is not explained there either, and also not needed (it can be introduced later on near the equation where it is defined). Similarly, the natural gradient formula in (6) is left unexplained.
- Instead of saying "some norm", it could be called as usual a p-norm and denoted ||..||_p.
- The readability is decreased by the use at some places of negative notations rather than positives that are easier to follow, and sometimes even easier to write. Example: tau does not tend to 0, instead of tau_k > 0 \forall k; or r(s, a) does not tend to +-\infty, instead of |r(s, a)| <= R_max which is easier to write. The statement "x < eps for all eps > 0" actually defines x = 0.
- I suggest introducing some different notation for the rescaled versions of the parameters w, such as w' or v, since this would make the change to the algorithm easier to follow.

Some other presentation comments:
- Perhaps the abstract might make explicit the fact that NAC must be modified to make it equivalent to OPI
- The keyword "reinforcement learning" could be added somewhere early in the introduction
- "In addition to that the class" doesn't seem to be a well-formed sentence
- Regarding terminology: in "non-optimistic" policy iteration, it may also be necessary to have w^_k be "complete" in the sense explained one para below, in addition to kappa = 1.
- "no modifications to the existing algorithms are needed" -- with the exception of the forgetting factor in NAC(kapppa), this should be pointed out
- The difference between the first and second part of Theorem 3 could be pointed out better/in advance, I had to reread the passage a couple of times to understand. Also, the progression from Theorem 2 to Theorem 3 might be explained in advance to prepare the reader.

AFTER AUTHOR FEEDBACK:
I am grateful to the authors for responding to my comments. My point about "no modifications" was not to say that we have new algorithms in the paper, but rather suggesting to admit that NAC and OPI are not immediately equivalent; before that some changes to NAC to make it more general must be made. I still think that saying "x < eps for all eps > 0" is mathematically equivalent to saying x=0 (in the context of positive x) so as the authors say, it would be good to revise the statement.

I keep my overall (positive) opinion about the paper.
Summary: An insightful paper about the convergence of optimistic policy iteration and its relationship to natural actor critic algorithms. It can make an impact in the approximate RL field, by pointing out a quite general approach that does not work, which suggests a possible way forward to something that does work. Some formalism can be introduced better, and making the dense discussions easier to read would also help.
Author Feedback

Author rebuttal: We thank the reviewers for their accurate and detailed reviews, and for updating their reviews after the rebuttal phase. We agree with many of the improvement possibilities noted by the reviewers and will incorporate them into the final version. The reviewers also asked some minor questions, to which we provide answers in the following.

w.p.1: with probability one (we will open up this in the paper)

'"no modifications to the existing algorithms are needed" -- well, with the exception of the forgetting factor in NAC(kappa)?': NAC(kappa) is equivalent to the unmodified form of optimistic policy iteration (OPI), so adding the forgetting factor does not lead to a new algorithm but merely redefines OPI using the NAC notation. An answer to further comments: Interpolation between the algorithms (with NAC being approached in the limit) can be performed also with unmodified OPI using its original (traditional) definition, with the implication that the unmodified, un-redefined NAC is directly a limiting special case of OPI. We do not claim that unmodified NAC and OPI would be /equivalent/, but that the former is a /limiting special case/ of the latter.

Non-optimistic policy iteration: yes, w_k indeed needs to be "complete" (will be corrected).

x < epsilon, for all epsilon > 0: The expression is analogous to the statement 'x \to 0' in a limit expression (cf. the epsilon-delta definition of limit), except for being applicable also outside a limit statement (i.e., without the presence of another variable that would go toward something while x goes toward zero). We have replaced it with a clearer alternative.

We decided to keep the "negative notations" in the paper, as replacing them with "positive" ones would either lose preciseness or add clutter.

An advance explanation of Theorem 3 was added to the extended version (there was no space for it in the length-limited version).